# Targeting Hypoxia-Inducible Factor-1α for the Management of Hepatocellular Carcinoma

**DOI:** 10.3390/cancers15102738

**Published:** 2023-05-12

**Authors:** Kenneth N. Huynh, Sriram Rao, Bradley Roth, Theodore Bryan, Dayantha M. Fernando, Farshid Dayyani, David Imagawa, Nadine Abi-Jaoudeh

**Affiliations:** 1Division of Interventional Radiology, Department of Radiological Sciences, University of California Irvine, Orange, CA 92868, USA; 2Division of Hematology and Oncology, Department of Medicine, Chao Family Comprehensive Cancer Center, University of California Irvine, Orange, CA 92868, USA; 3Division of Hepatobiliary and Pancreas Surgery, Department of Surgery, University of California Irvine, Orange, CA 92868, USA

**Keywords:** hypoxia, hypoxia-activated prodrugs, hepatocellular carcinoma, HIF-1α, tirapazamine, trans-arterial embolization

## Abstract

**Simple Summary:**

Hypoxia, or low oxygen levels within tumor tissue, presents a significant challenge for the management of hepatocellular carcinoma (HCC). Hypoxic microenvironments can induce changes in gene expression and cellular metabolism that promote tumor growth, metastasis, and resistance to chemotherapy and radiation therapy, making the cancer cells more aggressive and difficult to treat. Targeting hypoxia in HCC has emerged as a promising strategy for improving treatment outcomes. The aim of this review is to summarize the current knowledge on the role and biochemical pathways of hypoxia in the progression of HCC. This review also discusses the potential therapeutic options for targeting hypoxia for the management of HCC.

**Abstract:**

Hypoxia-inducible factor 1 alpha (HIF-1α) is a transcription factor that regulates the cellular response to hypoxia and is upregulated in all types of solid tumor, leading to tumor angiogenesis, growth, and resistance to therapy. Hepatocellular carcinoma (HCC) is a highly vascular tumor, as well as a hypoxic tumor, due to the liver being a relatively hypoxic environment compared to other organs. Trans-arterial chemoembolization (TACE) and trans-arterial embolization (TAE) are locoregional therapies that are part of the treatment guidelines for HCC but can also exacerbate hypoxia in tumors, as seen with HIF-1α upregulation post-hepatic embolization. Hypoxia-activated prodrugs (HAPs) are a novel class of anticancer agent that are selectively activated under hypoxic conditions, potentially allowing for the targeted treatment of hypoxic HCC. Early studies targeting hypoxia show promising results; however, further research is needed to understand the effects of HAPs in combination with embolization in the treatment of HCC. This review aims to summarize current knowledge on the role of hypoxia and HIF-1α in HCC, as well as the potential of HAPs and liver-directed embolization.

## 1. Introduction

Hepatocellular carcinoma (HCC) is the most common primary malignancy of the liver and is the sixth most common malignancy globally [1]. The incidence of HCC is rapidly increasing, and it is currently the third most common cause of cancer-related mortality [1,2]. Several treatment options have been established for HCC, and they are categorized as either curative-intent or palliation. Curative options include surgical resection, ablative therapy, potentially radiation segmentectomy, and liver transplantation. At the time of diagnosis, less than 30% of patients are eligible for curative therapy. Palliative options include trans-arterial embolization (TAE), radioembolization, targeted therapies as mono-therapies or in combination with immunotherapies, and radiation. Cytotoxic chemotherapy has a limited role in the treatment of HCC due to the underlying hepatic dysfunction and chemoresistance properties of HCC [3,4,5,6,7]. Despite advancements in diagnostic and therapeutic modalities, the prognosis for HCC patients remains poor, necessitating the identification of novel molecular targets for effective intervention [8].

Tumor hypoxia, or the inadequate oxygenation of solid tumors, is a common condition that occurs in all types of solid tumor, including HCC [9]. Recent studies have shown that hypoxia has been linked to poor prognosis and is a major contributor to the development and progression of HCC [10,11,12,13,14,15,16,17,18,19]. Hypoxia can have various causes, including decreased oxygen delivery to tissues, increased oxygen consumption by tissues, or a combination of both. Hypoxia leads to the activation of hypoxia-inducible factors (HIFs), which play a critical role in promoting tumor growth and angiogenesis [11,12,16,19]. HIF-1α, a subunit of HIF-1 and an integral transcription factor that modulates cellular responses to hypoxia, has emerged as a critical player in HCC progression [20]. Elevated HIF expression and serum HIF-1α levels have been associated with poor clinical outcomes, as well as reduced overall survival [20]. HIFs also contribute to drug resistance and the ability of cancer cells to invade and metastasize [13,14,16,17]. In addition, acute hypoxic cells are considered resistant to chemotherapy and radiation therapy, rendering these therapies ineffective at destroying cancerous cells [17,21,22]. This can contribute to the development and progression of HCC, making it a significant treatment challenge.

To combat the effects of hypoxia in HCC, various strategies for targeting hypoxia are being explored. These strategies include hypoxia-activated cytotoxic agents, HIF inhibitors, angiogenesis inhibitors, and hyperbaric oxygen therapy [23,24,25,26]. TAE has emerged as a crucial treatment modality for unresectable HCC; however, TAE-induced hypoxia may paradoxically promote HCC progression by upregulating HIFs. By leveraging this artificial amplification of hypoxia, the combination of TAE with hypoxia-activated prodrugs is currently being explored in early clinical trials [27,28]. In this review, we will describe the relationship between hypoxia and hepatocellular carcinoma, with a focus on the mechanisms by which hypoxia contributes to tumor progression and the potential therapeutic strategies that exploit tumor hypoxia using hypoxia-activated cytotoxic agents.

## 2. The Role of Hypoxia in Tumor Progression

Understanding the relationship between tumoral microenvironments and hypoxia is crucial to understanding the latter’s therapeutic implications in HCC. Tumoral microenvironments consist of physical and biochemical factors that interact with cancer cells and influence their behavior. These factors include the extracellular matrix, cytokines, growth factors, and oxygen tension [19,29,30,31,32]. The microenvironment within a tumor can vary greatly, with areas of high oxygen tension and low oxygen tension (hypoxia) existing simultaneously within the same tumor [29,30].

Due to its blood supply, the liver is one of the organs that is most vulnerable to hypoxia, as oxygenated blood from the hepatic artery provides 25% of the blood supply, and deoxygenated blood from the portal vein supplies the remaining 75%. The levels of influent and effluent oxygen tension in the hepatic sinusoids are 60–65 mmHg and 30–35 mmHg, respectively [29,30]. This contrasts with other tissue capillaries with oxygen tensions of 74–104 mmHg and 34–46 mmHg [29]. Additionally, there is an oxygen demand mismatch due to the relatively slow blood flow through the hepatic sinusoids and the relatively high metabolic rate of hepatocytes. This is exacerbated in cirrhotic patients, in whom hypoxemia ranges from 10–40%, depending on the degree of hepatic dysfunction [30]. As such, solid tumors such as HCC are exposed to three types of hypoxia: chronic diffusion hypoxia from oxygen demand mismatch, acute/intermittent perfusion hypoxia from abnormal vascular flow dynamics due to disorganized angiogenesis, and anemic hypoxia from hypoxemia in the setting of hepatic dysfunction [19]. This oxygen tension is pronounced with larger tumor size, with chronic diffusion hypoxia occurring when tumor cells are beyond 70 µm from the blood supply and cellular necrosis occurring when the distance expands beyond 180 µm [17,18,19,32].

Hypoxia plays a significant role in the development and progression of tumors. Tumors are characterized by rapid cell proliferation, which leads to an increased demand for oxygen. To meet this demand, tumoral angiogenesis ensues, with new blood vessels supplying the tumor. However, due to the disorganization of new blood vessel development, there are areas of the tumor that are not well-perfused with oxygen. These areas of hypoxia lead to a series of molecular adaptations that allow cancer cells to survive in low-oxygen environments. This results in the further growth and spread of tumors through the activation of HIFs, the promotion of genetic mutations in cancer cells, and the inhibition of cellular apoptosis, as well as resistance to chemotherapy and radiation therapy in acute hypoxic cells.

## 3. Molecular Pathways of Hypoxia

The molecular response to hypoxia is mediated by HIFs, which are activated in response to low oxygen levels [19,30,31,32]. HIFs are composed of two subunits: HIF-1α and HIF-1β. Under normoxic conditions, HIF-1α is hydroxylated by prolyl hydroxylase enzymes (PHDs) and marked for degradation by the ubiquitin–proteasome pathway [19,30,31,32]. However, under hypoxic conditions, the activity of PHDs is downregulated, allowing HIF-1α to accumulate within the cell nucleus and form a complex with HIF-1β [19,30,31,32]. This HIF complex binds to hypoxia-responsive elements (HREs) in the promoter regions of target genes, such as vascular endothelial growth factor (VEGF), tumor protein p53 (p53), and B-cell lymphoma 2 (Bcl-2), leading to transcriptional activation (Figure 1) [19,25,30,33,34].

HIFs play a role in the regulation of several genes that are important for cellular adaptation to hypoxia. HIFs promote the metabolic reprogramming of HCC from oxidative phosphorylation to anaerobic glycolysis through the activation of glucose transporters 1 and 3 (GLUT-1 and GLUT-3) (the Warburg effect), facilitating survival under hypoxic conditions [19,25]. Cellular proliferation, however, is slowed or halted in acute hypoxic conditions and grants cancer cell resistance to chemotherapy and radiotherapy, which target rapidly proliferating cells [19,25]. A preclinical study of HCC rat models reported a decrease in tumor cell proliferation in acute hypoxia [35]. However, as tumors outgrow their blood supply over time, chronic hypoxia develops, with paradoxical adaptive cellular responses through the stabilization of HIFs and the resumption of transcriptional activation of genes involved in angiogenesis, cellular proliferation, and cell survival [20,36,37]. The upregulation of HIF-1α and VEGF in the setting of chronic hypoxia was observed in a preclinical study of HCC rat models, resulting in tumor growth [38,39]. Additionally, upregulating the expression of anti-apoptotic genes, such as B-cell lymphoma 2 (Bcl-2), desensitizes HCC cells to apoptosis [19,25].

## 4. Significance of Serum HIF-1α Levels in the Biology of HCC

Serum HIF-1α is upregulated in HCC, with elevated levels associated with a poor prognosis [11,12,13,14,15,16]. High tissue HIF-1α levels are associated with increased microvessel density through the upregulation of VEGF and other angiogenic factors [25,33,40]. Several studies have shown that high HIF-1α levels are associated with larger tumor size, higher tumor grade, decreased overall survival, and resistance to chemotherapy and radiation therapy [11,12,13,14,15,16]. The overexpression of HIF-1α has also been observed to induce the epithelial-to-mesenchymal transition (EMT), enabling tumor cells to acquire a mesenchymal phenotype, which facilities invasion, migration, and dissemination from the primary tumor site [13,14,16,19,41]. Preclinical studies have reported the suppression of EMT and decreased metastatic potential of HCC cells through the inhibition of HIF-1α [11,12,13,14,15,16]. Targeting HIF-1α is a potential therapeutic strategy for HCC.

## 5. Intra-Vascular Liver-Directed Therapies for the Treatment of HCC

Trans-arterial liver-directed therapies (LDTs) are widely used therapeutic options for HCC [42]. LDTs include bland trans-arterial embolization (TAE), conventional trans-arterial chemoembolization (cTACE), drug-eluting embolic TACE (DEB-TACE), and trans-arterial radioembolization (TARE). TAE involves the infusion of embolic materials into tumor-feeding arteries to occlude the blood supply to the tumor, causing tumor necrosis and a reduction in tumor size. cTACE involves the infusion of lyophilized forms of one or more chemotherapeutic agents (e.g., doxorubicin, cisplatin, or mitomycin) mixed with Lipiodol into tumor-feeding arteries, followed by occlusion of the tumoral vasculature with embolic material such as beads, Gelfoam, or another embolic material. DEB-TACE involves the administration of chemotherapy-loaded beads, typically doxorubicin or idarubicin for HCC, into tumor-feeding arteries, resulting in occlusion of the feeding vasculature with gradual release of the drug over time. In TARE, microspheres impregnated with the radioisotope yttrium-90 (^90^Y) are selectively delivered to the tumor-feeding vasculature, where they emit beta rays to induce DNA damage and tumor cell death.

Most chemotherapeutic agents, such as doxorubicin, rely on the presence of oxygen to generate reactive oxygen species (ROS) and induce DNA damage, which is impeded in hypoxic environments. As such, studies have demonstrated no significant difference in overall survival among the TAE, cTACE with doxorubicin, or DEB-TACE embolization techniques [43,44,45,46]. Additionally, larger tumor size was associated with a lower rate of complete response, from 66% for HCC smaller than 4 cm to 25% in HCC greater than 5 cm [47]. Although still unclear, the success of TAE is proposed to be limited by the presence of hypoxia within the tumoral microenvironment, which can cause resistance to the treatment [48,49,50]. Hypoxia protects cancer stem cells from necrosis. Hypoxia-induced angiogenesis results in the disorganized formation of new blood vessels that can resupply the tumor with oxygen and nutrients, allowing cancerous cells to survive despite embolization [26,40,51]. The addition of embolotherapy exacerbates an existing hypoxic microenvironment by decreasing oxygen delivery to the tumor and further activation of the hypoxia signaling pathway (Figure 2) [52].

Increases in HIF-1α levels were observed after TAE compared to before TAE in rabbit VX2 models [15]. In vitro studies of tissues from patients with HCC led to the upregulated expression of HIF-1α and COX-2 after cTACE, which was associated with worse overall survival compared to patients without elevated HIF-1α and COX-2 levels [16]. Clinical studies have also observed increased serum HIF-1α and VEGF levels after cTACE, with levels peaking at 1–7 days before gradually decreasing post-cTACE [11,12,40,51,53]. Significantly lower serum HIF-1α and VEGF levels were also observed in patients who achieved a complete response compared to those who exhibited a partial response, stable disease, or progressive disease [11]. Notably, however, serum HIF-1α and VEGF levels do not return to baseline levels after cTACE [12]. The upregulation of HIF-1α can result in the persistent recurrence of the tumor post-embolization. To combat this, a combination of arterial occlusion with hypoxia-activated prodrugs (HAPs) has been proposed, with the rationale of inducing a hypoxic environment to enhance the cytotoxic effects of hypoxia-activated agents [27,28,54,55,56].

## 6. Therapeutic Strategies That Target Tumor Hypoxia

One approach to overcoming the effects of hypoxia in cancer treatment is the use of prodrugs, which are inactive compounds that are converted into active drugs via enzymatic or metabolic processes within the body. Unlike traditional chemotherapy, HAPs are advantageously designed to be activated specifically in hypoxic regions of the tumor, where they may be more effective at treating cancer cells while minimizing damage to healthy tissue. Additionally, HAPs may be able to bypass drug resistance mechanisms that are commonly associated with traditional chemotherapy [25,33]. Several classes of HAPs include nitroaryl-based prodrugs, quinones, aliphatic N-oxides, and aromatic bioreductive prodrugs [25,33]. The bioreductive prodrugs that have reached clinical trials for the treatment of HCC are further elaborated upon in this review (Table 1).

### 6.1. Aromatic Prodrugs

Tirapazamine (TPZ) is an aromatic prodrug that has been shown to have anti-tumoral activity for the treatment of HCC in preclinical and clinical studies [25,27,28,33,54]. TPZ is converted to an active form by the enzyme cytochrome P450 2B1 (CYP2B1), which is overexpressed in the hypoxic regions of tumors [25,33]. This active form of TPZ generates reactive oxygen species that lead to DNA damage and impaired DNA repair, leading to the death of cancer cells as well as cancer stem cells (Figure 3) [25,33]. In phase I and II clinical trials of TPZ for the treatment of cervical, head and neck, and non-small-cell lung cancer, treatment was well tolerated without significant toxicity and showed promising responses and overall survival rates [33]. However, in several phase III trials, the addition of TPZ to chemoradiation therapy regimens failed to show any response benefit [33,58,59,60]. This may be, in part, due to the administration of TPZ in the absence of sustained hypoxia. The combination of TAE with tirapazamine has been investigated as a potential strategy to overcome the resistance of HCC to TAE due to hypoxia. Intravenously administered TPZ with subsequent hepatic artery ligation (HAL) demonstrated near-complete tumor necrosis (>99%) of HCC in hepatitis B virus X protein transgenic mouse models, with the sparing of normal hepatocytes [54]. This was more effective than HAL alone or HAL with doxorubicin, which showed 0% and approximately 5% tumor necrosis, respectively [54]. In TPZ-treated mice with multiple HCC lesions, it was also observed that lesions supplied by the ligated hepatic artery underwent extensive necrosis, whereas those not supplied by the ligated hepatic artery remained intact [54]. These results confirm the effectiveness of TPZ in hypoxic microenvironments, as well as its superiority to doxorubicin.

Early clinical studies have tested the efficacy of HAPs in patients with HCC in combination with TAE. In a first-in-human phase I trial by Abi-Jaoudeh et al., the combination of TPZ with TAE in treatment-naïve patients with unresectable HCC achieved a 60% complete response rate and an 84% overall complete and partial response rate (per the modified Response Evaluation Criteria in Solid Tumors (mRECIST) guidelines), despite a mean tumor size of 6.53 cm ± 2.60 cm with a median of two lesions per patient (Figure 4) [27]. This contrasts with a 52% overall response rate from cTACE in a systematic review of 10,108 patients [48]. Additionally, no significant differences in response were observed between HCC lesions smaller and greater than 5 cm, suggesting durable effectiveness of the treatment in larger tumors [27]. Another phase I trial by Liu et al. demonstrated a 47% complete response rate and 65% overall response rate (per the mRECIST guidelines) in an Asian population with patients who had failed TACE [28]. There were no dose-limiting toxicities and no serious drug-related adverse events in either study [27,28]. In both trials, tumor oxygenation status before and after intervention was not evaluated, which may be an added consideration in future trials. The selection of patient populations that would benefit from TPZ with TAE, possibly those exhibiting high expressions of HIFs or severe tumor hypoxia, would also need to be carried out. Phase II trials investigating the efficacy of TPZ with TAE combined with nivolumab in advanced-stage HCC (NCT03259867), and comparing TPZ with TAE versus cTACE in intermediate-stage HCC, are currently ongoing (NCT03145558).

### 6.2. Nitro-Based Prodrugs

TH-302, or evofosfamide, is a nitro-based HAP that is activated by HIF-1α and induces DNA damage and apoptosis in cancer cells through the generation of ROS [24,25,61]. Preclinical studies have observed significant tumor cell death, the inhibition of tumor growth, and the increased survival of xenograft HCC mouse models with the administration of TH-302 [62,63]. The efficacy of TH-302 was also noted to be enhanced in hypobaric conditions and attenuated in hyperbaric conditions [63]. In another preclinical study using rabbit VX2 models, administration of the prodrug TH-302 in addition to cTACE demonstrated significantly smaller tumor volumes, lower growth rates, and higher necrotic fractions when compared to cTACE [55]. In a phase I clinical trial of 18 patients with HCC or renal cell carcinoma (RCC) treated with TH-302 combined with sorafenib, 55.6% achieved complete, partial, or stable disease per the RECIST 1.1 criteria [57].

PR-104, a 3,5-dinitrobenzamide-2-mustard, is also a nitro-based HAP that is activated under hypoxic conditions. Once activated, it can exert cytotoxic effects by acting as a DNA interstrand cross-linking agent and covalently bonding two strands of DNA, thereby preventing replication and transcription in rapidly dividing cells [25,33]. The utility of PR-104 in combination with sorafenib for the treatment of HCC was explored in a phase I trial; however, the study was discontinued due to dose-limiting toxicities related to thrombocytopenia and neutropenia, with no or only a partial response to therapy [64]. This was attributed to poor clearance of the activated PR-104A form due to impaired glucuronidation in patients with advanced HCC [64,65]. A preclinical study later demonstrated a significant reduction in tumor growth after the administration of PR-104 in HCC xenograft mouse models exhibiting normal glucuronidation [65]. Further investigations need to be conducted to determine the appropriate patient population for therapy with PR-104.

### 6.3. Aliphatic Prodrugs

AQ4N [1,4-bis{[2-(dimethylamino-N-oxide)ethyl]amino}-5,8-dihydroxyanthracene-9,10-dione] is an aliphatic N-oxide prodrug that is converted to an active AQ4 form under hypoxic conditions. Once active, AQ4 strongly binds to DNA with a 1000-fold cytotoxic effect compared to its inactive form through the inhibition of topoisomerase II [25,33]. Preclinical trials have shown slowed tumor growth using AQ4N in solid tumor-bearing murine models, especially when combined with radiation therapy [6,66]. However, there are no clinical trials investigating the efficacy of AQ4N in HCC to date.

### 6.4. Quinone Prodrugs

Mitomycin C (MMC) is a quinone-based prodrug that has been widely established in cancer treatments due to its DNA cross-linking properties and its prevention of replication and transcription. One retrospective study investigating MMC in combination with cTACE observed an overall response rate of 76%, with a 39.3% overall survival rate at 5 years [67].

While the MMC derivatives porfiromycin (POR) and apaziquone (EO9) have been developed to exhibit greater hypoxia selectivity, these prodrugs did not show superiority over MMC or lacked efficacy in the treatment of solid tumors in clinical trials [25,68,69]. Preclinical studies performed after these clinical trials attribute EO9’s lack of efficacy in treatment response to poor pharmacokinetics when systemically administered, whereas the direct intra-tumoral administration of EO9 improved anti-tumor activity, suggesting EO9’s possible application in locoregional therapy [25,69,70]. The efficacies of POR and EO9 in the treatment of HCC have yet to be investigated.

### 6.5. Other Strategies Targeting Tumor Hypoxia

Other developing therapeutic strategies for targeting tumor hypoxia include natural agents and metformin.

Various natural agents, such as curcumin, resveratrol, sanguinarine, and ginsenosides, have been shown to inhibit HIF-1α and VEGF expression, thereby suppressing tumor angiogenesis in hypoxic tumors [71,72,73,74,75]. While these natural agents have shown effectiveness in targeting hypoxia in preclinical models, the clinical translation of natural agents for targeting tumor hypoxia remains limited due to poor bioavailability profiles of these agents [76]. To overcome these barriers, various strategies have been employed, including the use of nanoparticles, liposomes, and prodrug formulations; however, well-designed clinical trials are needed to establish the safety, efficacy, and optimal dosing regimens of these natural agents [77,78].

Metformin, an antidiabetic drug, has gained attention as a potential adjunct therapy due to its pleiotropic effects on cancer cells, including the modulation of hypoxia [79]. Metformin’s anticancer properties have been attributed to both direct and indirect mechanisms. Its direct mechanisms include the suppression of HIF-1α and HIF-2α by promoting their proteasomal degradation [80]. Indirectly, metformin suppresses HIF-1α translation, and also reduces intracellular oxygen demand, leading to decreased HIF-1α protein levels and decreased stabilization of HIFs under hypoxic conditions [81,82]. Combining metformin with cTACE has shown promise in clinical studies, with improved overall survival and time to progression in metformin users compared to non-users [83,84]. While metformin is a promising adjunct therapy for HCC patients undergoing cTACE, randomized controlled trials are required to elucidate the optimal dosage and timing of metformin in combination with cTACE.

Anti-angiogenic therapy, immunotherapies, and targeted therapies are additional strategies for targeting tumor hypoxia. Anti-angiogenic therapies target the tumor vasculature and reduce intratumoral hypoxia. Immunotherapies accentuate the body’s immune response to target cancerous cells and have been shown to be effective in some types of cancer. Therapies targeting specific genetic mutations found in tumors have also been shown to be effective [52,85].

## 7. Conclusions

Targeting tumor hypoxia with HAPs, such as TPZ and TH-302, has shown early promise for the treatment of HCC. Tumor hypoxia can be further exploited with the addition of TAE to enhance the efficacy of treatment with HAPs. Future directions for these approaches include further exploring the safety and efficacy of these treatments. Identification of the patient population that would benefit the most from this approach would also need to be determined. Phase II trials investigating HAPs with TAE are currently underway. Combination therapies with HAPs and other HCC treatments, such as immune checkpoint inhibitors or targeted therapies, may also be explored to enhance the efficacy of treatment. Overall, the development of these novel therapies has the potential to greatly improve outcomes for patients with HCC, and continued research and innovation in this field will be crucial to advancing the field of cancer treatment.

## Figures and Tables

**Figure 1 cancers-15-02738-f001:**
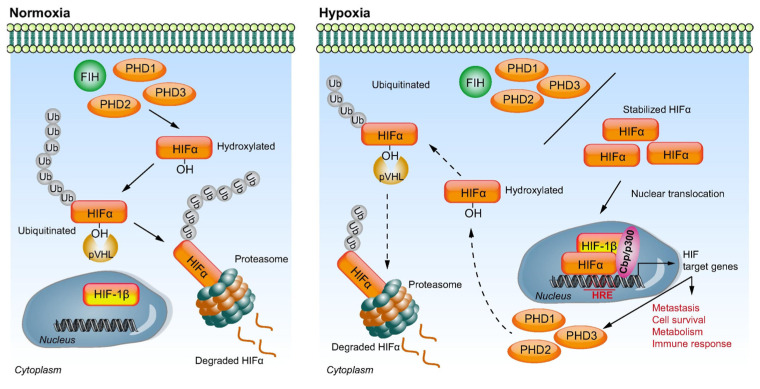
Schematic representation of hypoxia-inducible factor (HIF) signaling under differing oxygen tensions. Under normoxic conditions, HIF-1α is hydroxylated by prolyl hydroxylases (PHD1-3) and factor-inhibiting HIF (FIH). The hydroxylated HIF-1α is ubiquitinated by von Hippel–Lindau (pVHL) E3 ubiquitin ligase, resulting in proteasomal degradation. Under hypoxic conditions, PHD and FIH activity is inhibited. The stabilized HIF-1α is translocated into the nucleus, where it dimerizes with HIF-1β. This HIF complex interacts with co-activators Cbp/p300 and binds to hypoxia response elements (HREs), resulting in activation of the transcription of the HIF target genes involved in metastasis, cell survival, metabolism, and immune response. A self-regulating negative feedback loop eventually hydroxylates and ubiquitinates HIF-1α, with resultant degradation [34].

**Figure 2 cancers-15-02738-f002:**
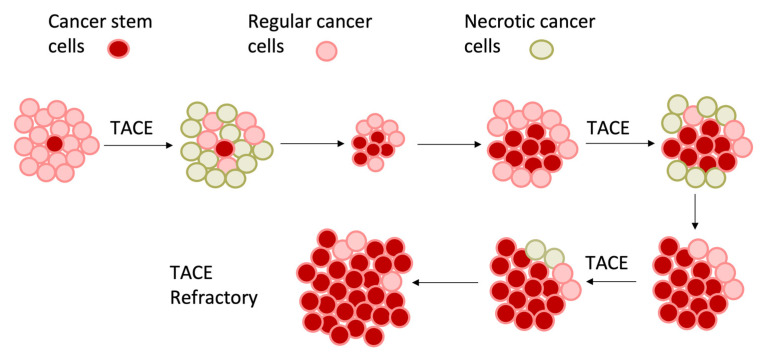
Schematic representation of effects of trans-arterial embolization on HCC with necrosis, with selection of hypoxia-resistant tumor cells and cancer stem cells leading to embolization refractoriness and failure.

**Figure 3 cancers-15-02738-f003:**
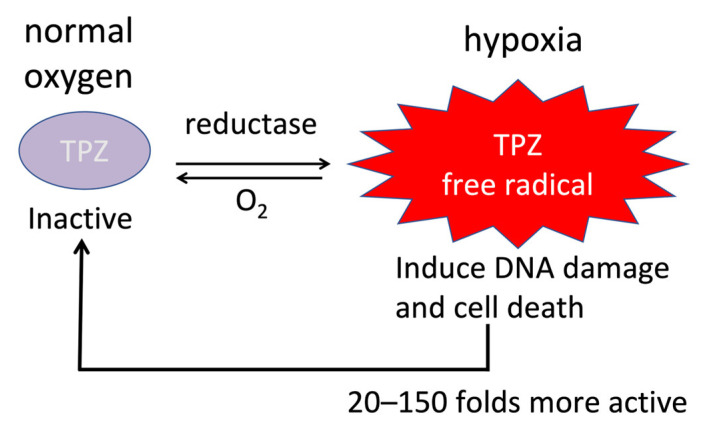
Mechanism of action of tirapazamine.

**Figure 4 cancers-15-02738-f004:**
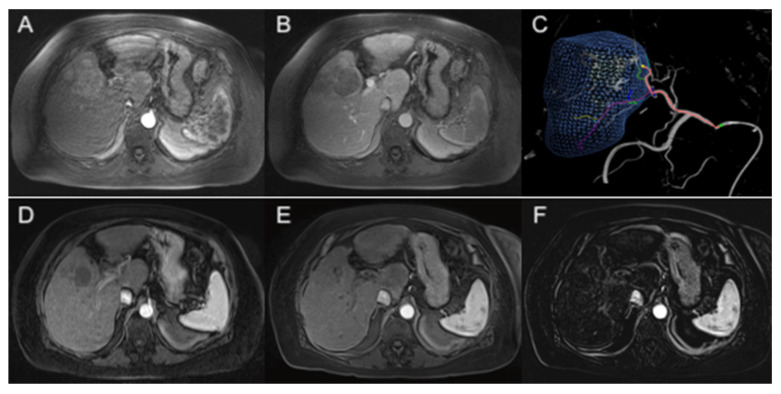
A 78-year-old female with biopsy-proven HCC and AFP 6307. (**A**) Initial arterial-phase axial MRI image demonstrates a large enhancing lesion measuring 68 mm in hepatic segment V. (**B**) Portal venous-phase axial MRI image shows the corresponding lesion with venous washout. (**C**) Intraprocedural cone-beam CT with navigation guidance software showing the segmented lesion (blue). (**D**) Arterial-phase axial MRI image obtained 2 months after embolization with TPZ demonstrates significant decrease in size of the nonviable segment V lesion with no enhancement, compatible with complete treatment response. (**E**,**F**) Arterial-phase and subtraction axial MRI images at 2.5 years shows continued decrease in size of the nonviable lesion with no enhancement, confirming durable complete response with AFP 4.1.

**Table 1 cancers-15-02738-t001:** Summary of selected preclinical and clinical trials using HAPs for the treatment of HCC.

Study	Phase of Trial	Cancer of Interest	Intervention	PFS	mOS	ORR	
Lin 2016 [54]	P	HCC in HBx transgenic mice	HAL alone DXR/HAL TPZ/HAL	-	-	-	>99% necrosis in TPZ/HAL ~5% necrosis in DXR/HAL No detectable necrosis in HAL alone
Duran 2017 [55]	P	VX2 tumor-bearing rabbits	TH-302 alone cTACE alone TH-302/cTACE	-	-	-	Higher necrotic fraction, tumor shrinkage, and lower tumor growth rate in TH-302/cTACE on day 14 compared to TH-302 or cTACE alone (*p* < 0.05)
Abi-Jaoudeh 2021 [27] (*n* = 27)	I	HCC	TPZ/TAE	80.5% at 6 months	52 months	84.0%	
Tran 2021 [57] (*n* = 18)	Ib	HCC (*n* = 12), RCC (*n* = 6)	TH-302/sorafenib	6.3 months (HCC)	13.9 months (HCC)	55.6%	
Liu 2022 [28] (*n* = 17)	I	HCC	TPZ/TAE	72.6% at 6 months	29.3 months	64.7%	

PFS—progression-free survival; mOS—median overall survival; ORR—overall response rate; P—preclinical; I—phase 1; Ib—phase 1b; HCC—hepatocellular carcinoma; HBx—hepatitis B virus; HAL—hepatic artery ligation; DXR—doxorubicin; TPZ—tirapazamine; cTACE—conventional trans-arterial embolization; TAE—trans-arterial embolization.

## Data Availability

Not applicable.

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
