# Peer review of "Targeting Hypoxia-Inducible Factor-1α for the Management of Hepatocellular Carcinoma"

_cancers, 2023, doi:10.3390/cancers15102738_

Round 1

Reviewer 1 Report

Journal of Cancers

Review Article;

The article entitled “Targeting Hypoxia in the Management of Hepatocellular Carcinoma”. The author reviewed Hypoxia-inducible factor 1 alpha (HIF-1α) that can regulates cellular response to hypoxia and is upregulated in most tumors, leading to tumor angiogenesis, growth, and resistance to therapy. Hepatocellular carcinoma is a highly vascular tumor as well as a hypoxic tumor due to the liver being a relatively hypoxic environment compared to other organs. Trans-arterial chemoembolization and trans-arterial embolization are locoregional therapies that are part of the treatment guidelines for Hepatocellular carcinoma but can also exacerbate hypoxia in tumors as seen with HIF-1α upregulation post hepatic embolization. Hypoxia-activated prodrugs are a novel class of anticancer agents that are selectively activated under hypoxic conditions, potentially allowing for targeted treatment of hypoxic Hepatocellular carcinoma. Research is needed to understand the effects of Hypoxia-activated prodrugs in combination with embolization in the treatment of Hepatocellular carcinoma. This study summarizes current knowledge on the role of hypoxia and HIF-1α in Hepatocellular carcinoma.

I carefully read the manuscript and found it is wonderful effort by the author to review HIF-1α regulator in Hepatocellular carcinoma. But there is some minor revision needs and fulfill the mistake which the author could have done during writing. The article could be considered for publication in the prestigious Cancers Journal.

Comments for Authors

Ø  Abstract section, (line number 22) “hypoxia and is upregulated in most tumors” the author needs to revise the statement. In most or in every tumor there is problem of hypoxic effect?

Ø  Revise the keyword and make it in one or two words.

Ø  “Introduction Section” the author needs to explain more the introduction section and include latest references.

Ø  The author needs to include natural agent’s effect against tumor hypoxia.

Ø  The author needs to include more references.

Ø  The author needs to explain in graphical image the effect of HIF-1a in normoxia as well as hypoxia level and its effect on molecular level.

Ø  Could the author explain the hypoxia effect on metastasis?

Cite the following references.

v  DOI: 10.2174/1871520622666220831124321

Author Response

Reviewer 1 Comments

  1. Abstract section, (line number 22) “hypoxia and is upregulated in most tumors” the author needs to revise the statement. In most or in every tumor there is problem of hypoxic effect?

This statement is revised. There is hypoxic effect seen in all types of solid tumors, with some more severely than others.

  1. Revise the keywordand make it in one or two words.

The keywords are revised and shortened to one to two words each.

  1. “Introduction Section” the author needs to explain more the introduction section and include latest references.

The introduction section was revised to provide a comprehensive overview of our review on targeting HIF-1α in the management of HCC.

  1. The author needs to include natural agent’s effect against tumor hypoxia.

Cite the following references.

v  DOI: 10.2174/1871520622666220831124321

Added section 6.5 “Other Strategies Targeting Tumor Hypoxia” with brief discussion of natural agents, metformin, anti-angiogenic therapy, immunotherapies, and targeted therapies. The effects of natural agents and metformin on HIF expression as well as the combination therapy of metformin and cTACE are briefly discussed. We appreciate the shared reference, with sanguinarine and other natural agents briefly discussed in the new section 6.5 “Other Strategies Targeting Tumor Hypoxia”.

  1. The author needs to include more references.

Additional references were included to support the information presented in this manuscript.

  1. The author needs to explain in graphical image the effect of HIF-1a in normoxia as well as hypoxia level and its effect on molecular level.

A graphical schematic demonstrating the molecular biology and effects of HIF-1α in normoxic and hypoxic environments is adapted from: Wilson GK, Tennant DA, McKeating JA. Hypoxia inducible factors in liver disease and hepatocellular carcinoma: Current understanding and future directions. J Hepatol 2014;61(6):1397-406. License and permission for reuse is obtained and will be included in the submission.

  1. Could the author explain the hypoxia effect on metastasis?

The effect of hypoxia on metastasis is briefly discussed, including epithelial-to-mesenchymal transition which facilitate the dissemination and metastasis of HCC cells. This transition is induced with overexpression of HIF-1α and suppressed with inhibition of HIF-1α.

Reviewer 2 Report

In this review, titled “Targeting Hypoxia in the Management of Hepatocellular Carcinoma” the authors summarized the understanding of hypoxia in HCC and included the most recent key studies focused on the targeting of hypoxia in HCC. In general, I found this review interesting but I would recommend the following changes that could improve it:

1.      The statement on line 57:  “In addition, hypoxic cells are resistant to chemotherapy and radiation therapy, rendering these therapies ineffective in destroying cancerous cells”. Please provide references to original papers that demonstrated this information. Is this in the case of acute or chronic hypoxic cells?

2.      Authors state in lines 122-123 that “Cellular proliferation, however, is slowed or halted in hypoxic conditions and grants cancer cell resistance to chemotherapy and radiotherapy which target rapidly proliferating cells“ Acute or chronic condition? Are there any studies (in HCC animal models for instance) demonstrating that, on the other hand, hypoxia increases cell proliferation in HCC?

3.      Concerning HIFs, there are three variants of the α subunit, i.e., HIF-1α, HIF-2α, and HIF-3α, and three paralogues of the β subunit, i.e., HIF-1β, HIF-2β, and HIF-3β. The cellular response to hypoxia is generally mediated by the transcription factors HIF-1 and HIF-2. As the HIF-2 plays a crucial role in the chronic hypoxia process and is a crucial upstream regulator of VEGF and Notch1 signalling. Therefore, the authors should include the relevant information about HIF-2 in HCC as well. Or if authors decide to focus specifically on the role of hypoxia and HIF-1α in HCC, they might include HIF-1α to the title.

4.      The chapter: “Significant of Serum HIF-1α Levels in the Biology of HCC” Is composed of the following sentences: “HIF-1α regulates the expression of genes involved in several cellular 115 processes, including angiogenesis, metabolism, and apoptosis, all of which are critical in the development and progression of HCC. Firstly, high tissue HIF-1α levels are associated 117 with increased microvessel density through the regulation of VEGF and other angiogenic factors [20,26,27]. Secondly, HIF-1α promotes the metabolic reprogramming of HCC from oxidative phosphorylation to anaerobic glycolysis by activation of glucose transporter 1 and 3 (GLUT-1 and GLUT-3), the Warburg effect, facilitating survival under hypoxic conditions [17,20]. Cellular proliferation, however, is slowed or halted in hypoxic conditions and grants cancer cell resistance to chemotherapy and radiotherapy which target rapidly proliferating cells [17,20]. Additionally, HIF-1α desensitizes HCC cells to apoptosis by upregulating the expression of anti-apoptotic genes, such as B-cell lymphoma 2 (Bcl-2) [17,20]. Overexpression of HIF-1α has also been observed to induce with epithelial-to-mesenchymal transition (EMT), enhancing the ability of invasion and migration of HCC cells [11,12,14,17]. Several studies have shown that high HIF-1α levels is associated with  larger tumor size, higher tumor grade, decreased overall survival, and resistance to chemotherapy and radiation therapy [9–14]. Targeting HIF-1α is a potential therapeutic strategy in HCC.”

The majority of the statements repeat previous statements, from “2. Hypoxia in Tumor Progression or 3. Molecular pathways of hypoxia” where the role of HIF is described. Additionally, the information in the stated paragraph is not linked to serum HIF1 levels and should be moved to Chapter 2 or 3. This needs to be revisited.

5.      As the title of the manuscript is: “Targeting Hypoxia in the Management of Hepatocellular Carcinoma”, authors can also comment on the role of metformin in HCC. In fact, metformin suppresses hypoxia-induced stabilization of HIFs through reprogramming of oxygen metabolism in HCC and it has been demonstrated that the metformin administration is associated with enhanced response to transarterial chemoembolization for HCC. Adding a chapter 6.5 called : other strategies targeting tumor hypoxia can also include the paragraph line 291-296.

6.      Hypoxia-activated prodrugs are active in severely hypoxic conditions. In light of the recent manuscript: "Cramer and Vaupel 2022 - Severe hypoxia is a typical characteristic of human hepatocellular carcinoma: Scientific fact or fallacy?", authors may include it in the discussion.

7.      Figure 3 presents the original new data of HCC case, which is surprising in the review paper. If data are linked to the previously published paper, would recommend adding the sentence in the figure legend: adapted from Abi-Jaoudeh et al, 2021. REF. In addition, please add an arrow or circle the area of the tumor for readers who are not familiar with reading MRI images.

8.      Table 1. “Summary of Selected Preclinical and Clinical Trials using HAPs in the Treatment of HCC.” The table does not summarise only clinical trials in HCC treatment but includes Cervical carcinoma, Head and neck. Please adjust as required.

9.      Concerning the clinical studies, was the tumour oxygenation status evaluated?  Are the treatments beneficial for the specific patient population (for instance for patients with a high HIFs expression / elevated tumour hypoxia)? Could that be the selection criterium before the treatment?

I noticed that minor editing is needed.

Author Response

Reviewer 2 Comments

  1. The statement on line 57:  “In addition, hypoxic cells are resistant to chemotherapy and radiation therapy, rendering these therapies ineffective in destroying cancerous cells”. Please provide references to original papers that demonstrated this information. Is this in the case of acute or chronic hypoxic cells?

    This statement is revised with the specification that resistance to chemotherapy and radiation therapy occurs predominantly in acute hypoxic cells, which exhibit decreased cellular proliferation. References are provided to support this statement.
  2. Authors state in lines 122-123 that “Cellular proliferation, however, is slowed or halted in hypoxic conditions and grants cancer cell resistance to chemotherapy and radiotherapy which target rapidly proliferating cells“ Acute or chronic condition? Are there any studies (in HCC animal models for instance) demonstrating that, on the other hand, hypoxia increases cell proliferation in HCC?

    The manuscript was revised to reflect the effects of hypoxia in both acute and chronic settings. Cellular proliferation is decreased in acute hypoxic cells. Cellular adaptive responses in the setting of chronic hypoxia led to the resumption of cellular proliferation, which was seen in a study of HCC rat models with resultant tumor growth.
  3. Concerning HIFs, there are three variants of the α subunit, i.e., HIF-1α, HIF-2α, and HIF-3α, and three paralogues of the β subunit, i.e., HIF-1β, HIF-2β, and HIF-3β. The cellular response to hypoxia is generally mediated by the transcription factors HIF-1 and HIF-2. As the HIF-2 plays a crucial role in the chronic hypoxia process and is a crucial upstream regulator of VEGF and Notch1 signalling. Therefore, the authors should include the relevant information about HIF-2 in HCC as well. Or if authors decide to focus specifically on the role of hypoxia and HIF-1α in HCC, they might include HIF-1α to the title.

    We appreciate this comment on the role of HIF-2 in chronic hypoxic environments and the implications it has in the pathophysiology of HCC. Both HIF-1 and HIF-2 certainly play crucial roles in the progression of HCC. HIF-1 has been extensively studied with human studies, whereas HIF-2 is less explored. The serum levels of HIF-1 have been studied in the setting of intervention with cTACE. In light of this, we have decided to emphasize the role of hypoxia and HIF-1α in HCC. This is change is reflected in the title of the manuscript.
  4. The chapter: “Significant of Serum HIF-1α Levels in the Biology of HCC” Is composed of the following sentences: “HIF-1α regulates the expression of genes involved in several cellular 115 processes, including angiogenesis, metabolism, and apoptosis, all of which are critical in the development and progression of HCC. Firstly, high tissue HIF-1α levels are associated 117 with increased microvessel density through the regulation of VEGF and other angiogenic factors [20,26,27]. Secondly, HIF-1α promotes the metabolic reprogramming of HCC from oxidative phosphorylation to anaerobic glycolysis by activation of glucose transporter 1 and 3 (GLUT-1 and GLUT-3), the Warburg effect, facilitating survival under hypoxic conditions [17,20]. Cellular proliferation, however, is slowed or halted in hypoxic conditions and grants cancer cell resistance to chemotherapy and radiotherapy which target rapidly proliferating cells [17,20]. Additionally, HIF-1α desensitizes HCC cells to apoptosis by upregulating the expression of anti-apoptotic genes, such as B-cell lymphoma 2 (Bcl-2) [17,20]. Overexpression of HIF-1α has also been observed to induce with epithelial-to-mesenchymal transition (EMT), enhancing the ability of invasion and migration of HCC cells [11,12,14,17]. Several studies have shown that high HIF-1α levels is associated with  larger tumor size, higher tumor grade, decreased overall survival, and resistance to chemotherapy and radiation therapy [9–14]. Targeting HIF-1α is a potential therapeutic strategy in HCC.”

The majority of the statements repeat previous statements, from “2. Hypoxia in Tumor Progression or 3. Molecular pathways of hypoxia” where the role of HIF is described. Additionally, the information in the stated paragraph is not linked to serum HIF1 levels and should be moved to Chapter 2 or 3. This needs to be revisited.

This chapter was noted to present repeated information seen in other sections of the text. This has been revised to minimize repeated statements. Additionally, the information not linked to serum HIF-1 levels were moved to their appropriate sections in the text.

  1. As the title of the manuscript is: “Targeting Hypoxia in the Management of Hepatocellular Carcinoma”, authors can also comment on the role of metformin in HCC. In fact, metformin suppresses hypoxia-induced stabilization of HIFs through reprogramming of oxygen metabolism in HCC and it has been demonstrated that the metformin administration is associated with enhanced response to transarterial chemoembolization for HCC. Adding a chapter 6.5 called : other strategies targeting tumor hypoxia can also include the paragraph line 291-296.

    Added section 6.5 “Other Strategies Targeting Tumor Hypoxia” with brief discussion of natural agents, metformin, anti-angiogenic therapy, immunotherapies, and targeted therapies. The effects of natural agents and metformin on HIF expression as well as the combination therapy of metformin and cTACE are briefly discussed.
  2. Hypoxia-activated prodrugs are active in severely hypoxic conditions. In light of the recent manuscript: "Cramer and Vaupel 2022 - Severe hypoxia is a typical characteristic of human hepatocellular carcinoma: Scientific fact or fallacy?", authors may include it in the discussion.

    We appreciate the reviewer’s comment in sharing this article. The viewpoints in this expert opinion by Cramer and Vaupel are certainly interesting. However, there are substantial preclinical and clinical studies that have extensively studied the effects of hypoxia in the progression of HCC. These studies are summarized in our review article of targeting HIF-1α in the management of HCC. We believe the viewpoints discussed in the expert opinion by Cramer and Vaupel will be better addressed with counterpoints in the form of a separate commentary or expert opinion to the referenced expert opinion.
  3. Figure 3 presents the original new data of HCC case, which is surprising in the review paper. If data are linked to the previously published paper, would recommend adding the sentence in the figure legend: adapted from Abi-Jaoudeh et al, 2021. REF. In addition, please add an arrow or circle the area of the tumor for readers who are not familiar with reading MRI images.

    The images used in Figure 3 in this article are original and were not taken/adapted from the published article by Abi-Jaoudeh et al, 2021. The case presented is one of many included in the dataset studied in the Phase I trial conducted at our institution by Abi-Jaoudeh et al, 2021. However, the images from this specific case were not used for any prior publication. The images in this figure have been annotated with circles and arrows to direct the readers to the lesion of interest.
  4. Table 1. “Summary of Selected Preclinical and Clinical Trials using HAPs in the Treatment of HCC.” The table does not summarise only clinical trials in HCC treatment but includes Cervical carcinoma, Head and neck. Please adjust as required.

    Rows presenting the trial results for cervical carcinoma and head/neck SCC have been removed from Table 1, in keeping with the presentation of trial data specific to HCC.
  5. Concerning the clinical studies, was the tumour oxygenation status evaluated?  Are the treatments beneficial for the specific patient population (for instance for patients with a high HIFs expression / elevated tumour hypoxia)? Could that be the selection criterium before the treatment?

    In both clinical studies using TPZ in the management of HCC, patient oxygenation status was evaluated (>92% SpO2) as a selection criterion, however, tumor oxygenation status was not evaluated as part of the trial designs in both clinical studies. This could be an added consideration for future trials with either invasive O2 measurements or indirectly measured with HIF1a serum levels, DCE-MRI with specific contrast probes, or hypoxia PET imaging.

Only phase 1 trials have been performed at this current time. The identification of patient populations that would most benefit from this treatment (those with high HIF expression or elevated tumor hypoxia) could be an added consideration for future phase 2/3 trials. 

Round 2

Reviewer 2 Report

The authors answered all comments.